# Cryo-electron tomography reveals the binding and release states of the major adhesion complex from *Mycoplasma genitalium*

**Lasse Sprankel, Margot P. Scheffer, Sina Manger, Utz H. Ermel, Achilleas S. Frangakis** [ID]*

Buchmann Institute for Molecular Life Sciences and Institute of Biophysics, Goethe University Frankfurt, Frankfurt, Germany

* achilleas.frangakis@biophysik.org

**Data Availability Statement:** Single particle cryo-electron microscopy densities were deposited in the Electron Microscopy Databank (EMDB) under the accession codes EMD-17590 for the canonical

## Abstract

The nap particle is an immunogenic surface adhesion complex from *Mycoplasma genitalium*. It is essential for motility and responsible for binding sialylated oligosaccharides on the surface of the host cell. The nap particle is composed of two P140-P110 heterodimers, the structure of which was recently solved. However, the interpretation of the mechanism by which the mycoplasma cells orchestrate adhesion remained challenging. Here, we provide cryo-electron tomography structures at ~11 Å resolution, which allow for the distinction between the bound and released state of the nap particle, displaying the *in vivo* conformational states. Fitting of the atomically resolved structures reveals that bound sialylated oligosaccharides are stabilized by both P110 and P140. Movement of the stalk domains allows for the transfer of conformational changes from the interior of the cell to the binding pocket, thus having the capability of an active release process. It is likely that the same mechanism can be transferred to other *Mycoplasma* species that belong to the pneumoniae cluster.

## Author summary

*Mycoplasma genitalium* is a sexually transmitted human pathogen that adheres to host epithelial cells thereby causing urethritis, cervicitis and pelvic inflammatory disease. Adhesion to host surfaces is critical for pathogenicity and is mainly mediated by an adhesion complex called the 'nap particle'. Previously the structure of the extracellular region of the nap particle was solved, which was in a non-adhering state. Here we use cryo-electron tomography of mycoplasma cells to capture the entire structure of the nap particles at different conformational states *in vivo*. In the tomograms, we see the interactions between the P140 and P110 proteins, the constituents of the nap particle, and how they synergistically create a pocket for binding and releasing sialylated oligosaccharides. Our results show that the intracellular regions anchor the tetrameric complex to allow the transition between the 'open' and 'closed' states. Altogether our work provides new insights into the elaborate adhesion mechanism of mycoplasma cells.

Nap, EMD-17589 for the expanded Nap, EMD-17587 for heterodimer class 1 and EMD-17588 for heterodimer class 2. The entire native Nap complex was deposited under the accession code EMD-17591 and the two distinct classes of the 'open' and 'closed' state of the native Nap with the accession codes EMD-17592 and EMD-17593, respectively. Coordinates of model building and MDFF have been deposited in the Protein Databank (PDB) under the accession codes 8PBX (heterodimer class 1), 8PBY (heterodimer class 2), 8PBZ (entire native Nap), 8PC0 (native Nap 'open' state) and 8PC1 (native Nap 'closed' state), respectively.

**Funding:** ASF was supported by the Deutsche Forschungsgemeinschaft, FR 1653/6-3 for LS, FR 1653/14-1 for UE and the Research Training Group iMOL GRK 2566/1 for MS and SM. The funders had no role in study design, data collection and analysis, decision to publish, or preparation of the manuscript.

**Competing interests:** The authors have declared that no competing interests exist.

## Introduction

*Mycoplasma genitalium* is a sexually transmitted pathogen associated with pelvic inflammatory disease [1] and has a genome size of 580 kb [2], and has thus been used as a minimal cell model in systems biology [3]. Despite the massively reduced genome and limited metabolic capabilities, mycoplasmas possess several immune evasion strategies such as molecular mimicry, antigenic variation, degradation of immune effector molecules, and negative regulation of the immune response [4,5]. In particular, antigenic variation of the most immunogenic proteins comprising the major adhesion complex has been shown to play a clinically relevant role [6]. A total of nine partial copies of the operon (4% of the genome) encoding the major adhesion complex have been identified in the mycoplasma genome, containing conserved and variable regions [7]. The latter have been demonstrated to be exchangeable by homologous recombination during infection [8]. These immune evasion strategies enable untreated mycoplasma infections to persist for several months or years and even manifest as chronic infections [9]. While mycoplasma are typically extracellular pathogens, they can also invade host cells, which is thought to be a critical step in chronification [9,10]. No approved vaccines are currently available [11], and given the natural resistance of mycoplasma to cell wall antibiotics, macrolides (antibiotics inhibiting protein synthesis) have been most commonly used to treat mycoplasma infections [12]. In the last years, however, an increased prevalence of macrolide-resistant mycoplasma infections has been reported [13]. It is therefore crucial to understand the infection and the associated adhesion mechanism.

Infectivity and cytadherence are mediated by a cell protrusion, the terminal organelle, which consists of the intracellular electron-dense core (EDC) and the surface-exposed adhesins, originally described as nap particles [14,15]. The intracellular EDC has been shown to be critical for the transmission of motility, while the nap particles, which surround the entire terminal organelle, are essential for adhesion and gliding motility [16,17]. Cryo-electron tomography (cryo-ET) of *M. genitalium* revealed the tetrameric organization of the nap particle complex, consisting of two P140-P110 heterodimers [18,19]. For gliding motility and cytadherence, sialylated oligosaccharides have been identified as being essential [20], and the binding pocket was identified in the crown region of P110 [21]. The P140-P110 heterodimers form the 'tight interface', while the 'loose interface' is formed between the two heterodimers. It has been proposed that when the tight interface is 'open' it can bind to sialylated oligosaccharides, while when the tight interface closes, the binding pocket is obstructed, which leads to the release of the sialylated oligosaccharide [19]. Further, it has been proposed that conformational changes of the intracellular domains of the nap particles communicate with the EDC, which allows for a coordination of the adhesion. To address the above hypothesis, it is crucial to capture the entire conformational cycle of the nap particle complex and thus the mechanism by which it passes information from the sialic acid binding pocket to the intracellular region of the complex.

Here, we determined the structure of the complete native nap particle complex *in-situ*, including the previously uncharacterized intracellular region at 11 Å resolution using cryo-ET. Classification showed an opening and closing of the tight interface accompanied by a repositioning of the stalks. These dynamics are only visible in cryo-electron tomography when the intracellular region is present. Additional in-depth analysis of the isolated nap particle complex by single-particle cryo-electron microscopy (cryo-EM) revealed distinct classes of the loose interface of the complex. Fitting of the atomically resolved structures reveals that bound sialylated oligosaccharides are stabilized by both P110 and P140.

## Results

### Classification of the isolated nap particle complex shows a closed tight interface and the loose interface in different conformations

To structurally characterize the nap particle complex, we performed single particle analysis of two preparations: (i) the isolated nap particle complex directly purified from *M. genitalium* and (ii) the isolated nap particle complex soaked with 6′-sialyl-lactose (6′-SL) after purification (Material and methods and S1 Fig). We obtained two datasets with 216,630 and 76,926 particles for each preparation, but only selected ~32% and ~65% of the particles that represented tetramers, respectively. Consistently for both preparations, the tight interface between P110 and P140 remains in the 'closed' state, where P140 occludes the sialic acid binding pocket of P110 (Fig 1c and 1d, **asterisk** and S2 Fig). Thus, despite having sialylated oligosaccharides in the original medium and soaking the purified protein with 6′-SL, an open state of the tight interface cannot be seen, as observed in tomographic data [19].

Classification of isolated nap particles revealed two predominant classes with different conformations of the loose interface, representing ~45% and ~26% of the population, with a resolution of 8.3 Å (EMD-17590; Fig 1a and 1c, and S3a Fig) and 7.3 Å (EMD-17589; Fig 1b and

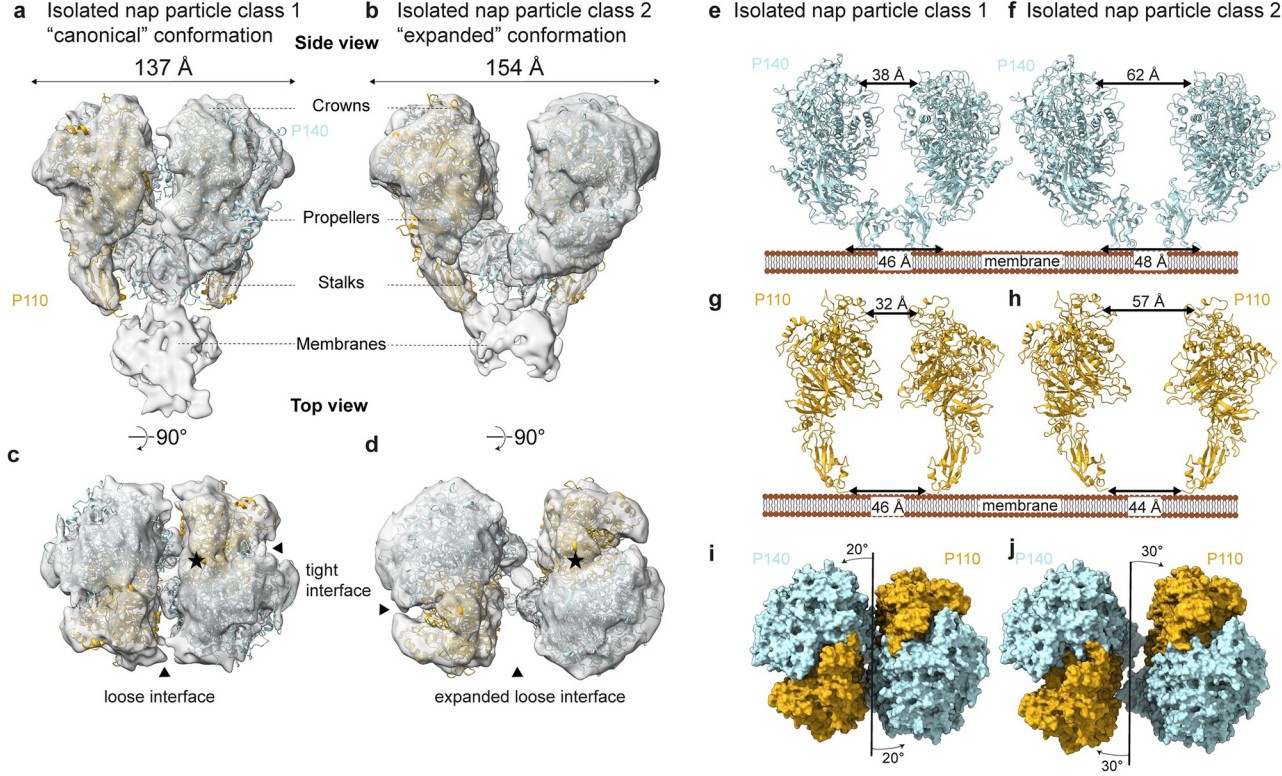

**Fig 1. Structural analysis of isolated nap particle complex by cryo-EM.** Loose interface and top view of the two main nap particle classes. The crystal structures of P110 (6R3T, yellow) and P140 (6S3U, light blue) were rigid-body fitted into the density maps. The loose interface view of **(a)** single particle canonical class, and **(b)** the expanded class shows the different angle between the two heterodimers that causes their distance to change. The locations of the crowns, propellers, stalks and membranes are indicated. Top views of **(c)** the canonical and **(d)** expanded class, show the widening of the loose interface, while the tight interface remains unchanged. In all single particle reconstructions and classes, the tight interface is in a 'closed' state, such that the binding pocket is occluded (indicated by a star). **(e,f)** Comparative distances between the stalk and crown regions of P140s with respect to one another. **(g,h)** Comparative distances between the stalk and crown regions of P110s with respect to one another. **(i,j)** A small change in the stalk position affects the angle between the heterodimers (10˚) and thus has an amplified effect on the crown positions at the loose interface.

1d, and S3b Fig), respectively. The P110 (6R3T) and P140 (6S3U) crystal structures were fitted into the density maps of both conformations and showed a discrete distance change between the heterodimers. In the first conformation the distance between the two heterodimers is 137 Å as measured between the Ser285 residues of P140 and resembles the distance between the two heterodimers as previously seen in the cryo-ET structure (EMD-10259) (hereafter the 'canonical conformation') (Fig 1a). In the second conformation, namely the 'expanded conformation', the distance between the two heterodimers is 154 Å (Fig 1b). The nap particles soaked with 6′-SL were solved at a resolution of 7.7 Å (EMD-18414; S3c Fig) and resembled the canonical conformation. When analyzing the arrangement of the P140s with respect to one another, the distance between the stalks for the expanded conformation is 2 Å wider apart compared to canonical conformation, but for the crown regions of P140, a difference of 24 Å can be measured (Fig 1e and 1f). By contrast, when analyzing the arrangement of the P110s with respect to one another, the distance between the stalks in the canonical conformation is slightly greater compared to the expanded conformation, and in the crown regions of P110, a distance increase of 25 Å can be measured (Fig 1g and 1h). These distance changes are caused by a change in angle of 10° between the heterodimers from the symmetry axis of the nap particle (Fig 1i and 1j). In both datasets, namely soaked with 6′-SL and unsoaked, the isolated nap particle complex has an unstructured intracellular region, and no density can be resolved, even when a focus refinement is performed. Thus, the intracellular and transmembrane regions do not play a role in the stabilization or dynamics of the complex in its purified state.

## Classification of the native nap particle from mycoplasma cells shows the tight interface in both the 'open' and the 'closed' state

*M. genitalium* ghost cells were prepared for cryo-electron tomography (cryo-ET) to preserve the nap particles in their native state [18]. Importantly, the mycoplasma cells are grown in complete medium where sialylated oligosaccharides are present. In the cryo-ET reconstructions, the entire nap particle including the intracellular domain can be seen, displaying a highly electron dense and structured region (S4b Fig, **white arrow**). A total of 36,720 nap particles were manually selected in the tomograms and subjected to sub-tomogram averaging. The structure of the entire native nap particle was resolved at a resolution of 11 Å (8PBZ; EMD-17591; S3h Fig). In contrast to previous studies, this analysis by classification of the native nap particle displayed both the 'open' and the 'closed' state of the tight interface, resolved at 17 Å and 18 Å, respectively (Fig 2, S3g and S3f Fig and S1 and S2 Movies). Although the resolution obtained is insufficient to resolve the sialylated oligosaccharides in the crown region, the fact that we obtain classes with the 'open' conformation is conclusive evidence for the bound state. Thereby, for the 'open' state of the nap particle (8PC0, EMD-17592), the crystal structure of P110 bound to 6′-SL (6R43) as well as the crystal structure of P140 (6RUT) were rigid-body fitted into the density map of the native nap particle. The fitting was subsequently refined to resolve minor steric clashes at the tight interface using molecular dynamics flexible fitting (MDFF). The resulting structure of the P110-P140 interaction site in the crown region for the 'open' state shows that the distance between P110 and P140 at the tight interface is sufficiently large so that the interfering loop (P140 residues 807–827) no longer interferes with the β-hairpin of P110 (residues 455–465) (Fig 2a, 2b and 2f). The structure indicates that P140 may additionally stabilize the binding to sialylated oligosaccharides by forming hydrogen bonds with residues from region 631–634 and loop 870–877 of P140, such as those between residues N634, S871 and 6′-SL (Fig 2b). For the 'closed' conformation (8PC1, EMD-17593), the fitting was performed in a similar manner using the P110 crystal structure in the unbound state (6R3T), showing the sialic acid binding pocket to be occluded by P140 by forming several

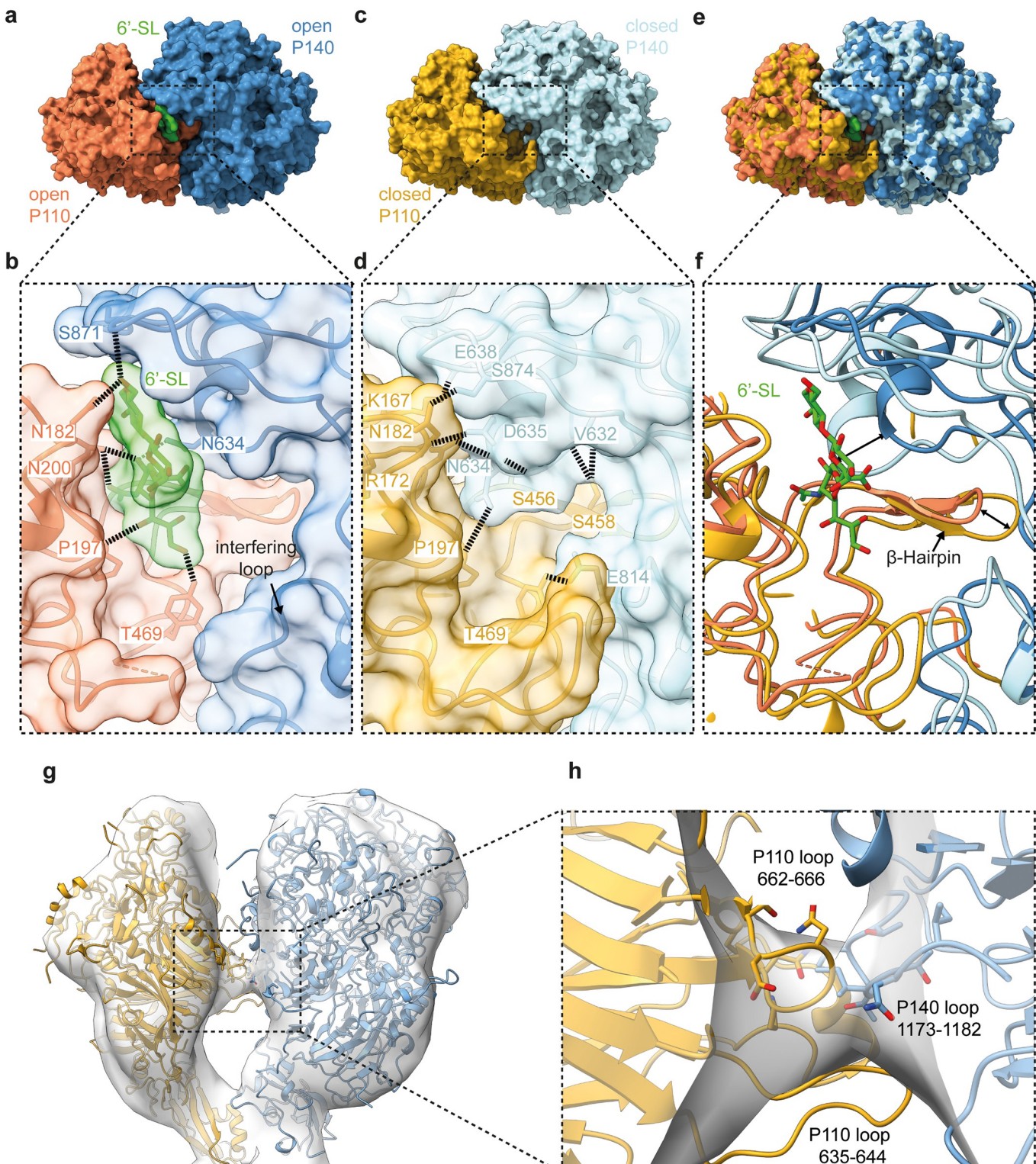

**Fig 2. Comparison between the open and closed conformation of the native nap particle tight-interface from the cryo-ET data.** Molecular dynamics flexible fitting of the P110 crystal structure bound to 6′-SL (6R43) combined with the crystal structure of P140 (6RUT) into the open tight interface of the native nap particle. For the closed cryo-ET conformation, MDFF was carried out using the P110 crystal structure in the unbound state (6R3T). (**a**) Top view of the surface representation of the open conformation of the P140 (blue)-P110(orange) heterodimer with 6′-SL (green) as ligand. (**b**) Enlarged view of the binding pocket bound to 6′-SL. The binding of 6′-SL is stabilized by hydrogen bonds from both P110 and P140. (**c**) Top view of the surface representation of the closed conformation of the P140 (blue)-P110 (yellow) heterodimer. (**d**) Enlarged view of the binding pocket illustrates hydrogen bonds formed between the P110 and P140. In this state, binding of 6′-SL is not possible. (**e**) Superimposition of the sialic acid binding pocket between the surface representation of the two different

open and closed cryo-ET classes (coloring identical to **b** & **d**). (**f**) Enlarged view of the superimposition shows the β-hairpin consistent with the closed and open conformation. (**g**) Side view of the loose interface showing electron density between the two heterodimers in the 'open' state. (**f**) Enlarged view of the loose interface connection showing an interaction between loop 662–666 of P110 and loop 1173–1179 of P140.

hydrogen bonds with P110, and thus closing the binding pocket (Fig 2c and 2d). The β-hairpin is in a conformation such that the binding of 6′-SL would cause steric clashes.

Importantly, the opening and closing of the tight interface is accompanied by an interaction between P110 and P140 at the loose interface of the nap particle (Fig 2g and S1 Movie). In the 'closed' conformation, there is no density seen at the loose interface, whereas in the 'open' conformation, P110 and P140 come into proximity at the loose interface, thus indicating an interaction between loop 662–666 of P110 and loop 1173–1179 of P140 (Fig 2h).

### The P140 stalks adopt different conformations but do not suffice to change the crown arrangement to an open conformation

We next used the single particle cryo-EM dataset of the purified nap particles to analyze the C-terminal stalks that connect the crown region with the sialic acid binding pocket to the trans-membrane helices and the intracellular domain (S1d Fig and S1 Table). For this, we analyzed 290,179 heterodimers which allowed the resolving of both stalk domains which had not been possible before, since previous heterodimer constructs were truncated. After classification, two distinct classes showing both the stalk regions of P110 and P140 (Fig 3a and 3b) were solved at resolutions of 3.3 Å and 3.7 Å, respectively (S3d and S3e Fig). The most dominant class (8PBX, EMD-17587) matched the crystal structure of the truncated heterodimer (6RUT), including the P110 stalk (6R3T). The second most dominant class (8PBY, EMD-17588) however, exhibited a different conformation of the P140 stalk region near the loop Ile1259-Leu1267 (Fig 3b). Rebuilding of the P140 stalk region resulted in a new model for P140. This showed that the loop region Ile1259-Leu1267 is shifted by approximately 10 Å towards the mycoplasma membrane (Fig 3b), which results in a movement of the residues Gly1241-Asp1245, previously named the 'hinge' region. Changes in the 'hinge' region have an immediate effect on the positioning of the crown region of the nap particle (Fig 3, additional density is indicated). Despite the substantially different arrangement of the P140 stalk, the overall conformation of the tight interface is in the 'closed' state, with the sialic acid binding pocket of P110 occluded by P140, indicating that flexibility in the P140 stalk alone does not trigger conformational changes at the tight interface. This agrees with previous surface plasmon resonance experiments, which showed that sialylated compounds 3′-SL and 6′-SL (which bind to P110 alone) do not bind to the P140-P110 heterodimer in solution [19].

### Tetrameric organization and intracellular domains are required for the 'open' conformation

To assess whether the movement of the stalks in the tetramer is sufficient to generate the 'open' conformation of the tight interface, we compared the position of the stalks of the native nap particles to the stalks of the isolated nap particle, with and without soaking with 6′-SL (Fig 4a and 4b). In both preparations of the isolated nap particles, the P110 stalk shows a larger movement in comparison to the native P110 stalk, presumably because the intracellular domain is unstructured, and the cytoplasmic C-terminal domains of the nap particle do not interact. In the native nap particle, the stalks move substantially less, however, minute changes at the stalks can have an amplified effect at the crown regions. In fact, classes showing an opening and closing of the tight interface are always associated with a repositioning of the P110

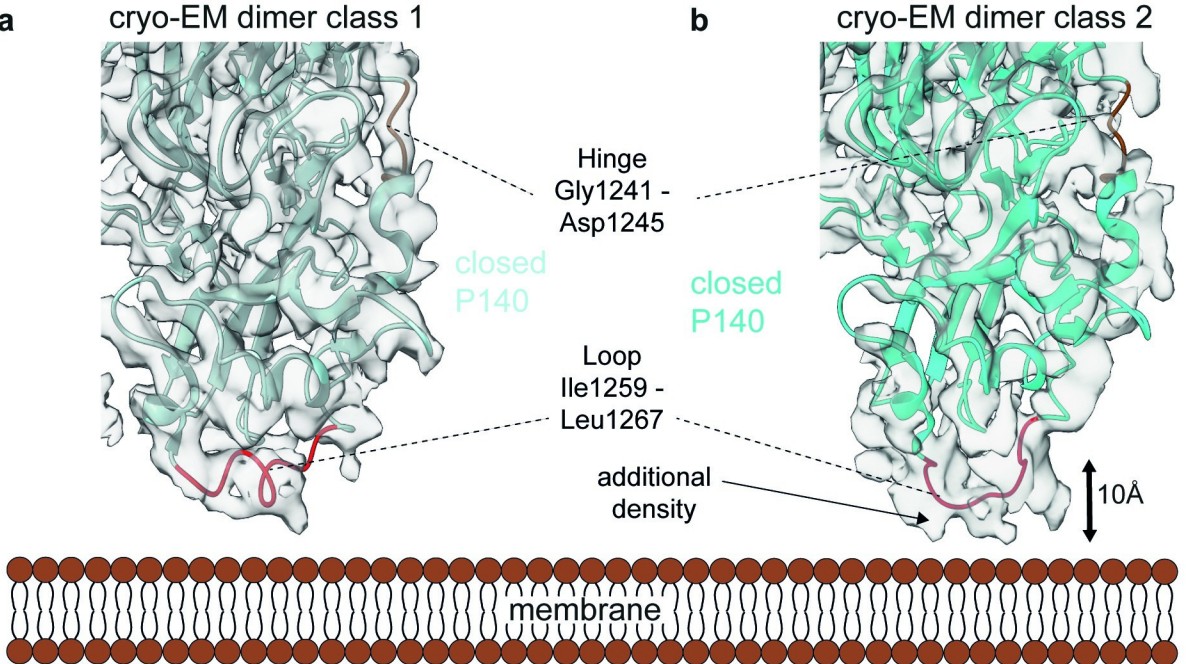

**Fig 3. Structure and conformational flexibility of the P140 stalks from the isolated nap particle heterodimer. (a)** The canonical purified nap particle heterodimer at 3.3 Å agrees with the crystal and **(b)** the alternative conformation of the P140 stalk at 3.7 Å displays substantial variations in the P140 stalk region. Indicated is the hinge loop Gly1241-Asp1245 (colored in brown), which connects the stalk to the propeller of P140, the loop Ile1259-Leu1267 (colored in red), which displays flexibility between the classes, and the additional density in the dimer class 2. The 10 Å shift of the loop Ile1259-Leu1267 is indicated by an arrow.

stalk. An additional 13° tilt of P110 is observed, which leads to its movement towards P140 at the loose interface. Also, the P140 stalk appears more rigid in the native nap particle classes, allowing for stabilization of the tetramer (Fig 4b and 4c). We therefore conclude that stable intracellular domains are needed for achieving both conformations.

## The intracellular region of the native nap particle is a compact four-helix bundle

The structure of the entire native nap particle was resolved at a resolution of 11 Å according to the gold standard criterion (8PBZ, EMD-17591) (Fig 5 and S3h Fig). The previously uncharacterized intracellular region appears compact proximal to the membrane and forms two distinct branches distal to the membrane. The P110 and P140 intracellular regions of the C-termini are predicted to be 87 amino acids (9.4kDa), and 67 amino acids (7.3 kDa), respectively, due to the anticipated length of the transmembrane helices. The previously uncharacterized residues 937–1053 of P110 and 1348–1444 of P140 were predicted with high confidence using Alphafold [22] as a four-helix bundle splitting into two-helix branches. The Alphafold model could be fitted into the density of the intracellular region using rigid body docking (Fig 5). The transmembrane region forms a ~40-residue four-helix bundle proximal to the membrane and an unstructured flexible loop (predicted with low confidence and therefore not displayed) distal to the membrane of approximately 36 and 28 amino acids for P110 and P140, respectively. The intracellular density measures ~90 Å along the symmetry axis of the nap particle, being ~50 Å wide at the distal end, with 5 Å between the two branches. The density allows for the docking of all four intracellular helices, showing that residues from P110 are likely to form the distal

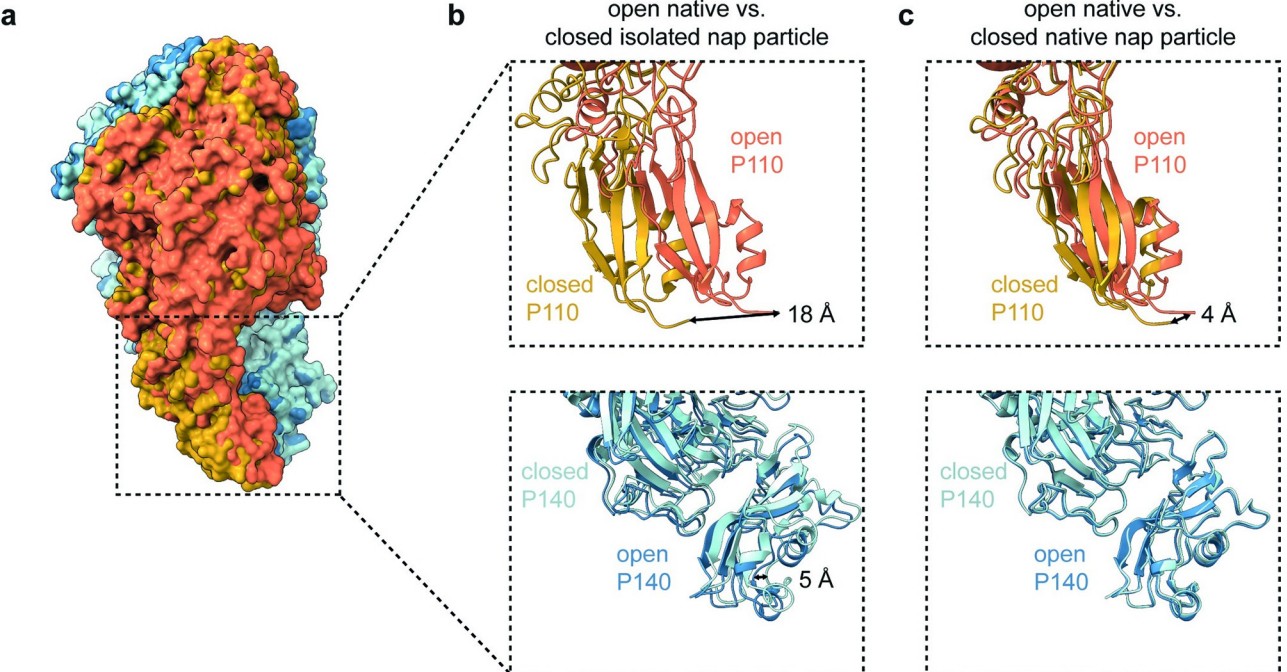

**Fig 4. Comparison between the P110 and P140 stalk positions in the 'open' and 'closed' conformations of the nap particle tight interface. (a)** Side view of the surface representation of the superimposition between the open and closed conformation of the native P140-P110 heterodimer from cryo-ET. **(b)** Ribbon representations of the different P110 and P140 stalk positions fitted into the open native nap particle cryo-ET density compared to the closed isolated nap particle density map. **(c)** Ribbon representations of the different P110 and P140 stalk positions fitted into the open native nap particle density map in comparison to the closed native nap particle density map. The movement of the stalks is much larger in the isolated nap particle since the intracellular domains are unstructured and the stalks are not anchored. Small differences in the stalk positions of the native nap particle can be sufficient for an opening and closing of the tight interface because of amplified movement at the crown region. Distances for P110 were measured at the C-terminal end of P110 (residue 936). For P140 the distance was measured at residue 1256.

branch of the intracellular density (Fig 5). Despite classification efforts, the intracellular region does not display differences between the 'open' and 'closed' state.

## Discussion

The nap particle is the major adhesion complex of mycoplasmas in the pneumoniae cluster and is crucial for the infectivity and cytadherence of the organism [16]. To evade the antibody response of the immune system, mycoplasmas elicit antigenic variations in the crown region of the nap particle, especially around the sialic acid binding pocket located on the P110 protein [4,23].

In this study, classification of the isolated nap particles showed that a small difference in the distance between the stalk domains translates into a large difference in the crown region, thus an amplified movement can be seen (EMD-17589, EMD-17590). A deviation of the angle between the heterodimers from the symmetry axis of the nap particle of 10° causes an amplification of the spacing between heterodimers at the loose interface (Fig 1i and 1j).

The structure of the entire native nap particle by cryo-ET depicts the 'open' and 'closed' state in the cellular context, which is not the case for purified nap particles. The density of the previously unresolved intracellular region was solved at 11 Å resolution, thereby providing a model of the entire nap particle. Unfortunately, the resolution of the cryo-ET data is insufficient to visualize the sialylated oligosaccharides, however in the 'open' state, the distance

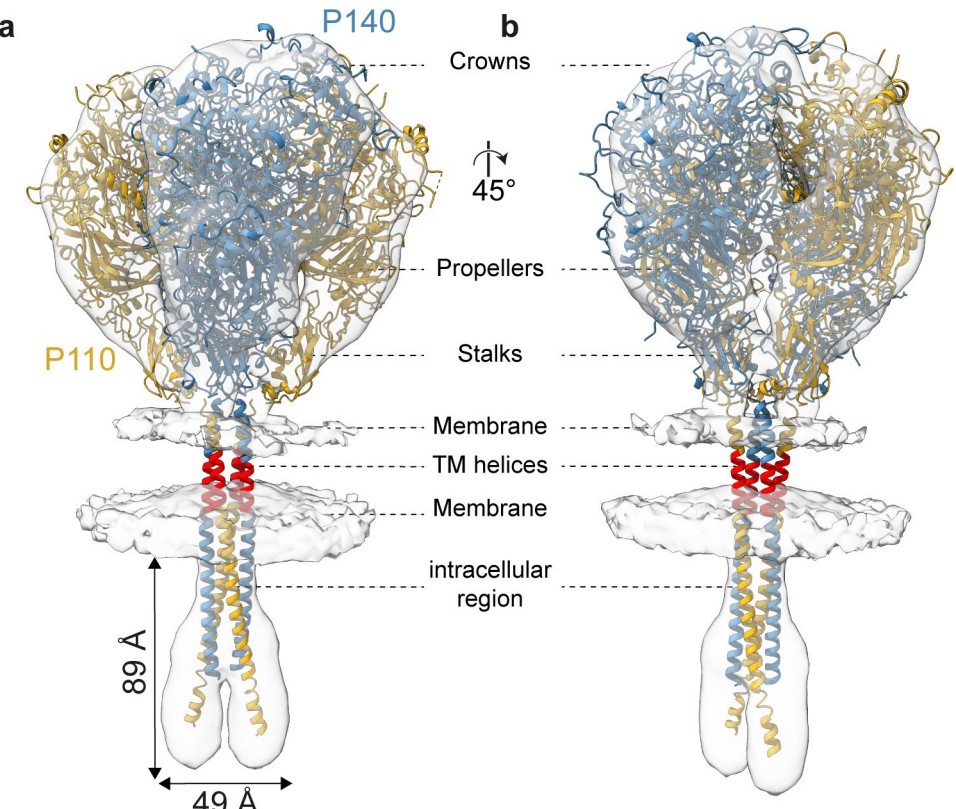

**Fig 5. Cryo-ET structure of the nap particle adhesion complex in *M. genitalium*.** The cryo-ET density map of the whole nap particle solved at 11 Å resolution shown as transparent white surface. The open conformation of the nap particle (8PC0) was fitted into the density. The previously uncharacterized residues 937–1053 of P110 and residues 1348–1444 of P140 were predicted as a four-helix bundle using Alphafold [22] and fitted into the density of the intracellular region. The residues 1017–1053 of P110 and 1416–1444 of P140 were predicted with little confidence as an unstructured loop and are not displayed in this model.

between P110 and P140 at the tight interface is sufficiently large that the interfering loop no longer interferes with the β-hairpin of P110. Importantly, the model (8PC0) shows that although P110 binds 6′-SL, P140 additionally stabilizes the binding pocket by forming hydrogen bonds to the sialylated oligosaccharide. In the 'closed' conformation, the sialic acid binding pocket is occluded by P140, which forms several hydrogen bonds with P110 and thus closes the binding pocket (8PC1).

The opening and closing of the tight interface is accompanied by an interaction between P110 and P140 at the loose interface of the nap particle (Fig 2g and S2 Movie). In the 'open' conformation (8PC0), the P110 and P140 come into proximity at the loose interface, thus indicating an interaction between loop 662–666 of P110 and loop 1173–1179 of P140 (Fig 2h). Consistent with these different conformational states, the in-cell crosslinking network of *M. pneumoniae* [24] confirms several interactions sites between the interfaces (S3 Table), which could be transferred by structural homology to the *M. genitalium* nap particle (S5 Fig) and likely can be transferred to other *Mycoplasma* species belonging to the pneumoniae cluster. In this regard, loop 635–644 of P110 does not contribute to the electron density at the loose interface but presumably moves outwards to form connections with the beta propeller region of P140, as suggested by the crosslink formed between Lys1289 and Lys818 in *Mycoplasma pneumoniae* (Fig 2h and S5 Fig and S3 Table).

The C-terminal stalks of P110 and P140 are known to be conserved among the different species within the pneumoniae cluster and antibodies directed against the stalks of the corresponding orthologs in *M. pneumoniae* inhibit adhesion [25]. It has therefore been proposed that the stalks are critical for the proper functioning of the nap particle [7,26]. Previously, it was hypothesized that a repositioning of the stalks enables a conformational change within the tight interface of the heterodimer, thereby influencing the binding of sialic acid by switching between an 'open' and 'closed' state [19]. High resolution cryo-EM reconstructions of the P140-P110 heterodimer (8PBX, 8PBY) suggest that the conformation of the P140 stalk in particular is critical for the functioning of the complex, in agreement with a previous study reporting an alternative conformation of the P140 stalk [27]. Here, a 10 Å shift of the Ile1259-Leu1267 loop towards the mycoplasma membrane affects the regions downstream and upstream of the loop at the P140 stalk interface (Fig 3) (8PBY). This is likely to induce the conformational change at the loose interface of the complex, which ultimately triggers the opening of the tight interface, by inducing the repositioning of the P110 stalks (Fig 6 and S2 Movie). The absence of heterodimers on the cell surface, as observed in cryo-ET data of mycoplasma cells, suggests that the tetrameric interaction is essential for proper functioning of the nap particle.

The conformation of the stalks of the native nap particles were compared to the stalks from single particle data of the isolated nap particle where the intracellular region is unstructured (Fig 4a–4c). In the isolated nap particle, the P110 stalk shows a large movement, accompanied by an additional 13° tilt of P110, in comparison to the stalk movement of the native nap particle. Because the intracellular regions are unstructured in the isolated nap particle, the stalk region can be displaced because the intracellular domain is not anchored. When the stalk is anchored through intracellular interactions, the displacement of the stalk is rather transferred to a large movement at the crown, causing an opening of the binding pocket. In classes of the native nap particle, a repositioning of the P110 stalk is associated with the opening and closing of the tight interface, which leads to its movement towards P140 at the loose interface. The P140 stalk is more rigid in the native nap particle classes, allowing stabilization of the tetramer.

The previously uncharacterized intracellular region appears compact proximal to the membrane and forms two distinct branches distal to the membrane in which the Alphafold model of a ~40-residue four-helix bundle splitting into two-helix branches could be fitted (Fig 5) (8PBZ, EMD-17591). The docking shows that residues from P110 are likely to form the distal branch of the intracellular density, and we hypothesize that the P110 intracellular helices exhibit a greater flexibility than those of P140, as indicated by differing stalk positions of P110 between the different classes. Although crosslinking mass spectrometry of the *M. pneumoniae* analogue for P110 shows five possible self-crosslinks and another ten cross-links with P140 in the intracellular regions, the intracellular region of P110 is not shown to interact with any other cytoskeleton proteins. Cross-linking mass spectrometry of the *M. pneumoniae* analogue for P140, on the other hand, shows that it has two self-crosslinks and two additional crosslinks with the cytoskeleton proteins HMW1 and P32 (P30 in *M. pneumonia*) [24].

## Materials and methods

### Sample preparation for cryo-electron tomography

*M. genitalium* G37 cells (ATCC 33530) were grown in cell culture flasks containing SP4 medium and incubated at 37 °C and 50% CO2 for 48 to 72 h. Surface-attached cells were harvested using a cell scraper and resuspended in 1 ml of fresh SP4 medium. Subsequently, 75 µl of the cells were inoculated in sterile 35 mm petri dishes with 3 ml SP4 medium containing 30 s glow discharged R1.2/1.3 Quantifoil grids (Quantifoil Micro Tools GmbH, Großlöbichau,

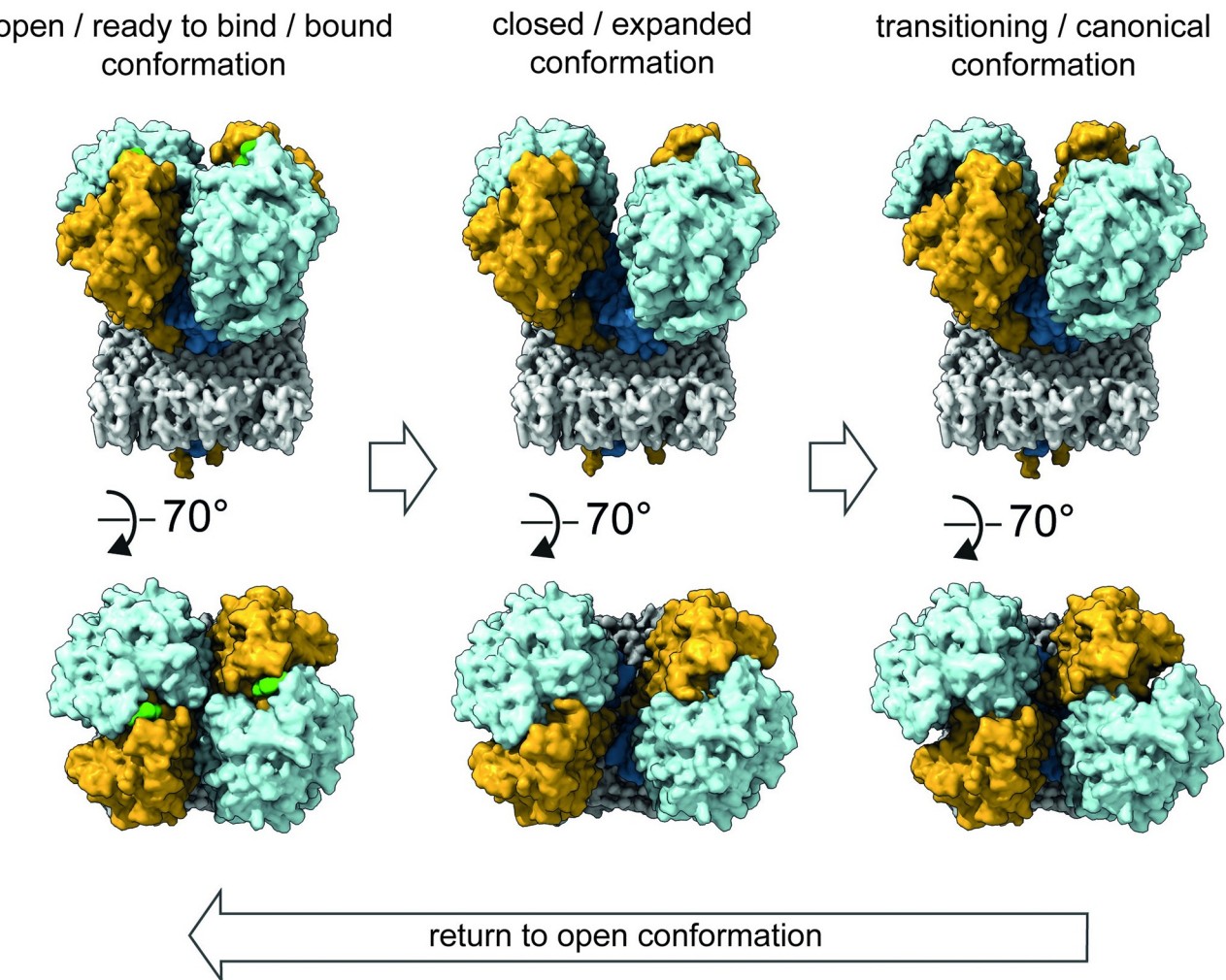

open / ready to bind / bound
conformation

closed / expanded
conformation

transitioning / canonical
conformation

↻ −70°          ↻ −70°          ↻ −70°

return to open conformation

**Fig 6. Model of the nap particle attachment mechanism.** In the open state, nap particle is bound to sialylated oligosaccharides mediated by P110 and further stabilized by P140. In addition, there is an interaction between P110 and P140 at the loose interface. Intracellular signaling, likely at the intracellular region of P140, then triggers a conformational change within the P140 stalks, changing the angle of the heterodimers relative to each other and thus transitioning to the expanded conformation where P140 and P110 do not interact at the loose interface. This allows P110 to reposition the stalk and close the tight interface, resulting in the "closed" conformation where no sialylated oligosaccharides can be bound.

Germany) and grown for 24 h. Directly before freezing, cells were treated for 1 min with 20 mM TEA pH 7.5, 0.5 M KCl, and 1% (v/v) Triton X-100 to generate ghost cells and subsequently washed three times in PBS. Finally, a 3.5 µl drop of 1:10 diluted fiducial markers (Protein A conjugated to 5 nm colloidal gold; Cell biology department, University Medical Center Utrecht, The Netherlands) was applied onto the grid, which was subsequently plunge-frozen in liquid ethane (Vitrobot Mark IV, Thermo Scientific, Waltham, USA) at 100% relative humidity, 4 °C, nominal blot force –1, with a blotting time of 8 to 12 s.

### Data collection of cryo-electron tomography data

Tilt-series were recorded using SerialEM v3.8 [28] at a nominal magnification of 105,000x (0.65 Å per pixel in super-resolution mode) in nanoprobe EFTEM mode at 300 kV with a Titan Krios (Thermo Scientific, Waltham, USA) electron microscope equipped with a GIF

Quantum S.E. post-column energy filter in zero loss peak mode and a K2 Summit detector (Gatan Inc., Pleasanton, USA). The total dose per tomogram was 120 e⁻/Å $^2$ using a frame rate of 0.1 s. The tilt-series covered an angular range from -60˚ to 60˚ with an angular increment of 3˚, recorded with a dose symmetric tilt scheme [29] and the defocus set to -3 μm (S2 Table).

## Cryo-electron tomography image processing

Tomograms were reconstructed using super-sampling SART [30] with a 3D CTF correction [31]. For volume visualization and isosurface rendering, the EMpackage [32] in Amira (Thermo Scientific, Waltham, USA) was used. Nap particles were manually selected and pre-aligned according to their respective 3D orientation within the tomogram. Firstly, the particles were averaged for several iterations with a full angular sampling for psi and phi while constraining theta to 45˚. Subsequently, the determined particle positions and orientations were placed back into the tomographic reconstruction using the EMpackage [32] in Amira (Thermo Scientific, Waltham, USA). Clearly misaligned particles were classified as bad particles and manually excluded from further processing. For final processing, a total of 36,720 particles were used and further averaged with gold standard averaging using the Artiatomi software package (github.com/uermel/Artiatomi). For the classification a multi-reference alignment was used providing three input references, both cryo-EM reconstructions of the tetramer and the previously published nap particle adhesion complex (EMD-10259), that were lowpass filtered to 40 Å. The two most prominent classes of the open and closed conformation contained 9,689 and 13,870 particles, respectively. All figures were rendered using the ChimeraX software package [33,34] with the ArtiaX plugin for analyzing and visualizing tomography data [35].

## Sample preparation and data collection for cryo-electron microscopy

To obtain nap particles with 6′-SL, the purified nap particle (25 μg/ml) was incubated with 10 mM 6′-SL in 75 mM Tris-HCl, pH 7.4, 400 mM NaCl, 5% glycerol and 0.5% octylglucoside detergent, for 30 min at room temperature prior to freezing. A 3.5 μL drop was then applied to a 45 s glow-discharged R1.2/R1.3 C-flat grid (Electron Microscopy Science), and plunge-frozen in liquid ethane (Vitrobot Mark IV, Thermo Scientific) at 100% relative humidity, 4˚C, a nominal blot force of -3, and a wait time of 10 s, with a blotting time of 10 s. Before freezing, Whatman 595 filter papers were incubated for 1 hour in the Vitrobot chamber at 100% relative humidity and 4 ˚C.

Dose-fractionated movies of the nap particle soaked with 6′-SL were collected with EPU v.3.1.3 (Thermo Fischer Scientific) at a nominal magnification of x105,000 (0.837 Å per pixel) in nanoprobe EFTEM mode at 300 kV with a Titan Krios G3i electron microscope (Thermo Scientific), equipped with a BioQuantum-K3 imaging filter (Gatan), operated in zero loss peak mode with 30 eV energy slit width. In total, 7,394 micrographs with 50 frames and a frame time of 0.05 s were collected. The K3 camera was operated in counting mode with a dose rate of ~15 electrons per $Å^2s^{-1}$, resulting in a total dose of 48 electrons per $Å^2s^{-1}$. Defocus values ranged from -1 to -4 μm.

## Single-particle cryo-electron microscopy image processing

CryoSPARC v3.2 [36] was used to extensively process the single-particle cryo-EM data from our previous publication [20]. For the processing of the tetramer with 6′-SL, cryoSPARC v4.2 was used. Beam-induced motion correction and CTF estimation were performed using CryoSPARC's own implementation. Particles were initially clicked with the Blob picker using a particle diameter of 200 to 300Å. Particles were then subjected to unsupervised 2D classification.

For the final processing of the dimers, the generated 2D averages were taken as templates for automated particle picking. For the final processing of the tetramer without 6′-SL, templates were generated from a de-novo 3D reference. For the tetramer with 6′-SL the results from the blob picker were used for the final processing. In total, 1,237,922 and 729,158 particles were picked and extracted with a box size of 300 pixels for the tetramer without 6′-SL and heterodimer, respectively. For the tetramer with 6′-SL 702,386 particles were picked and extracted with down sampled box size of 208 pixels. False-positive picks were removed by two rounds of unsupervised 2D classification. The remaining 216,630 particles (tetramer without 6′-SL), 76,926 particles (tetramer with 6′-SL) and 290,179 (heterodimer) were used to generate an *ab-initio* reconstruction with three classes followed by a subsequent heterogeneous refinement with three classes. For the processing of the tetramer without 6′-SL, the *ab-initio* reconstruction was skipped in the final processing, as the 3D reference was already generated previously. Instead, a heterogenous refinement, with the heterodimer and tetramer as input references, was used to separate the tetramer. For the final processing, 69,142 particles for the tetramer without 6′-SL, 50,318 particles for the tetramer with 6′-SL and 290,179 particles for the heterodimer were used. For the heterodimer, the beam-induced specimen movement was corrected locally. The CTF was refined per group on the fly within the non-uniform refinement for the heterodimer. For the nap particle tetramer without 6′-SL and the heterodimer, another heterogenous refinement yielded two different conformational states with a total of 18,408 particles (cryo-EM nap particle class 1 with a final resolution of 8.3 Å), 31,616 particles (cryo-EM nap particle class 2 with a final resolution of 7.3 Å), 149,542 particles (cryo-EM dimer class 1 with a final resolution of 3.3 Å), and 67,091 particles (cryo-EM dimer class 2 with a final resolution of 3.7 Å). For the tetramer with 6′-SL only one conformational state was observed at a final resolution of 7.7 Å. For the final processing of the nap particle tetramer C2 symmetry was applied (S1 Fig and S1 Table). Resolution was calculated according to the gold standard criterion of the FSC at 0.143 (S3 Fig).

## Model building, refinement and flexible fitting

For model building of the heterodimer, the crystal structure of the heterodimer (6RUT) was combined with the missing C-terminal domain of P110 (6R3T) using Coot [37]. Subsequently, the model was polished by alternating cycles of refinement using the "Real Space" protocol in the program Phenix [38,39] and manual reinterpretation and rebuilding with Coot using the cryo-EM dimer class 1 as reference for building. The generated model was used as a template for the remodeling of the C-terminal domain of P140 using the cryo-EM dimer class 2 as a reference. The model was again polished by alternating cycles of refinement in Phenix and rebuilding with Coot. For the molecular dynamics flexible fitting (MDFF) the Isolde plugin of ChimeraX was used [40]. For the 'open' state of the nap particle, the crystal structure of P110 bound to 6′-SL (6R43) as well as the crystal structure of P140 (6RUT) were first rigid-body fitted into the density map of the native nap particle. Subsequently, minor steric clashes at the tight interface were resolved using MDFF. The MDFF was performed at a temperature of 20 K with the highest simulation fidelity while applying simple distance restraints for P140 and restraining the positions of P110 on the pdb structure as a reference model. For the nap particle soaked with 6′-SL, the fitting was performed in a similar manner to the 'open' state using the identical starting models. For the 'closed' conformation, the fitting was performed in a similar manner, however, using the P110 crystal structure in the unbound state (6R3T).

## Supporting information

**S1 Table. Refinement and validation statistics for the P110/P140 heterodimer.**
(DOCX)

**S2 Table. Cryo-ET and cryo-EM data collection and processing parameters.**
(DOCX)

**S3 Table. Cross-linked residues between P1 and P40/P90 [21] and the corresponding residues of P140 and P110 based on structural homology.**
(DOCX)

**S1 Fig. Overview of the single particle processing of the nap particle complex. (a)** Representative micrograph and 2D classes of the nap particle complex. Local resolution map of **(b)** cryo-EM dimer class 1 and **(c)** cryo-EM dimer class 2 illustrates the poorer local resolution of the stalks. **(d)** Schematic overview of the cryo-EM processing workflow to obtain the different nap particle classes.
(TIF)

**S2 Fig. Comparison of the MDFF of the 6′-SL-soaked nap particle with the 'open' and 'closed' states from cryo-ET. (a)** Comparison of the MDFF of the 6′-SL-soaked nap particle complex with the MDFF of the 'open' state (8PC0) found in the cryo-ET data. Both MDFFs were performed in a similar manner using the identical starting models. Aligning the models on P110 shows, that the P140s are shifted by about 13 Å. Comparing the model of the 6′-SL-soaked nap particle with the **(b)** 'closed' model from the cryo-EM heterodimer (8PBX), no larger shift can be measured. It can be concluded that the 6′-SL-soaked nap particle resembles the 'closed' state.
(TIF)

**S3 Fig. Fourier shell correlations of different nap particle reconstructions.** Fourier shell correlation of the **(a)** single particle canonical nap particle, **(b)** single particle expanded nap particle, **(c)** single particle nap particle soaked with 6′-SL, **(d)** cryo-EM dimer Class 1, **(e)** cryo-EM dimer Class 2, **(f)** 'closed' state of the native nap particles, **(g)** 'open' state of the native nap particles, **(e)** native nap particle. Resolution was calculated according to the gold standard criterion of FSC 0.143.
(TIF)

**S4 Fig. Representative ghost cell of *M. genitalium*. (a)** Slice through a tomographic reconstruction of an *M. genitalium* ghost cell, which shows the intracellular terminal organelles, the surface exposed nap particles, the bacterial membrane and filaments attached to the terminal organelle. **(b)** Zoomed in region of nap particle complexes as a side view, showing the extracellular (black arrowhead) and intracellular region (white arrowhead) of the complex. **(c)** Zoomed in region of nap particle complexes as a top view. Scalebar of **(b)** and **(c)** is 20 nm.
(TIF)

**S5 Fig. The structure of the native nap particle with the estimated cross-linking positions. (a,b)** For the *M. pnuemoniae* nap particle, the N-domain of P40/P90 (6RJ1) and P1 (6RC9) were fitted into the extracellular density of the cryo-ET nap particle from *M. genitalium*. Cross-linking data from *M. pneumoniae* [21] suggests multiple interactions (colored in red) of the nap particle complex within the (a) loose and (b) tight interface. **(c,d)** For the *M. genitalium* nap particle, P110 (6R3T) and P140 (6S3U) were fitted into the extracellular density of the *M. genitalium* cryo-ET nap particle. The interactions (colored in red) correspond to the *M. pneumoniae* residues at the **(c)** loose and **(d)** tight interface. P110 and the N-domain of P40/

P90 are shown in yellow and P1 and P140 are shown in blue.
(TIF)

**S1 Movie. Alternating orthoslices through the cryo-ET structures of the nap particle adhesion complex in the 'open' (EMD-17592) versus 'closed' (EMD-17593) state.** The first set of orthoslices show the difference in the crown region where monomers come in close contact at the loose interface in the 'open' state, and a connection is formed. The second set of orthoslices show the difference in the propeller region which shows the conformational change and movement of the heterodimers with respect to one another. The last set of orthoslices shows the difference between the stalk regions in the 'open' and 'closed' state. Notably the differences from the crown to the stalk become smaller.
(MP4)

**S2 Movie. Model of the attachment mechanism of the nap particle.** Surface views of each step of the nap particle's attachment mechanism. The first set shows the attachment mechanism at the loose interface, the second set as a top view. In the open state, nap particle is bound to sialylated oligosaccharides mediated by P110 and further stabilized by P140. In addition, there is an interaction between P110 and P140 at the loose interface. Intracellular signaling, likely at the intracellular region of P140, then triggers a conformational change within the P140 stalks, changing the angle of the heterodimers relative to each other and thus transitioning to the expanded conformation. In this state, P140 and P110 no longer interact at the loose interface, allowing P110 to reposition the stalk and close the tight interface, resulting in the 'closed' conformation where sialylated oligosaccharides cannot be bound. Upon binding of sialylated oligosaccharides or intracellular signaling, the complex reverts to the 'open' state, allowing the cell to adhere.
(MP4)

## Acknowledgments

The authors thank Trevor Sewell for his advice for the modelling of the complex. We thank the Central Electron Microscopy Facility at the MPI of Biophysics in Frankfurt, which enabled us to collect the dataset of the tetramer soaked with 6′-SL. We thank in particular Sonja Welsch who assisted during the data collection. We thank David Aparicio for the purification of the isolated nap particle.

## Author Contributions

**Conceptualization:** Lasse Sprankel, Margot P. Scheffer, Achilleas S. Frangakis.

**Data curation:** Lasse Sprankel, Margot P. Scheffer, Utz H. Ermel.

**Investigation:** Lasse Sprankel, Sina Manger.

**Resources:** Achilleas S. Frangakis.

**Supervision:** Margot P. Scheffer, Achilleas S. Frangakis.

**Visualization:** Lasse Sprankel, Margot P. Scheffer, Achilleas S. Frangakis.

**Writing – original draft:** Lasse Sprankel, Margot P. Scheffer, Achilleas S. Frangakis.

**Writing – review & editing:** Lasse Sprankel, Margot P. Scheffer, Achilleas S. Frangakis.

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
