## [Decision Letter · Decision Letter 0]

24 Jul 2023

Dear Dr. Frangakis,

Thank you very much for submitting your manuscript "Cryo-electron tomography reveals the binding and release states of the native Nap adhesion complex" for consideration at PLOS Pathogens. As with all papers reviewed by the journal, your manuscript was reviewed by members of the editorial board and by independent reviewers. In light of the reviews (below this email), we would like to invite the resubmission of a significantly-revised version that takes into account the reviewers' comments.

Three reviewers and I have evaluated the manuscript, and as you can see, there was considerable disagreement about whether the work represents a sufficiently great advance to warrant publication in PLOS Pathogens. I find that it potentially does, but there are some important matters to address concerning the description of the results and their interpretation. I agree with Reviewer 1 that although these static images could relate to the mechanism of motility, there isn't sufficient evidence to be quite as conclusive about that connection as written in the Discussion section. Overall, in addition to addressing the suggestions of the reviewers and my own comments below, the limitations of this work (including the absence of substrate in the structures) should be made explicitly clear, and how this work is a significant advance from other published work on both the M. genitalium and M. pneumoniae structures should be described. With all of that laid out carefully, it will become much clearer whether this work reaches the bar for publication in PLOS Pathogens.

The word "nap" has been misused in recent manuscripts on the adherence complexes of Mycoplasma genitalium and Mycoplasma pneumoniae, and your next paper - whether it's this one or another - is a great opportunity to rectify the usage. "Nap" was never meant to stand in as the name of an adherence-associated complex of proteins. The word "nap," which was first used in regard to M. pneumoniae in 1982 (Hu et al., Science, 216:313-315), comes from the textile industry. It means "the raised hairs, threads, or similar small projections on the surface of fabric or suede" (from Oxford Languages). It is a collective noun, like "dirt." It refers to the appearance of the apparently raised material on the surface of the attachment organelle. Just like for dirt, we wouldn't call a single particle of dirt "a Dirt." We would call it "a dirt particle." If "P140/P110 complex" isn't sufficiently concise, I recommend calling this individual tetramer a "nap particle" - all lowercase, as proteins, chemicals, and other non-living biochemical entities are not capitalized in English.

One of the reviewers had these suggestions to me, which I'll pass along to you:

• 4 times a comma is missing, either upstream or downstream of “respectively“, pls also check commas when starting with “Subsequently“ or “Initially“

• Missing blanks at “Figure 1 a&c“/“Figure 1 g,h“..(no &?)/“Figure 2g“/“Figure 2h“

• lys1289 and lys818 should read Lys1289 and Lys818

• “the pneumoniae cluster“ should read “the pneumoniae cluster“ (no species->no italics)

• “..inhibit adhesion (Vizarraga 2020)“ should read “..inhibit adhesion (26).“

• “CO2“ should read “CO2“

• “Großlöblichau“ should read “Großlöbichau“

• “Clearly misaligned particles…were removed for further processing“: really? Or were they instead excluded from further processing?

• “…box size of 300 pixel“ should read “…box size of 300 pixels“

And I'll add some of my own:

• First line page 6 – change “are slightly wider apart” to “is slightly greater”

• Next to last paragraph of discussion – P30 is M. pneumoniae; P32 is M. genitalium

• Make sure you've written "M. pneumoniae" and not "M. pneumonia" in all instances

We cannot make any decision about publication until we have seen the revised manuscript and your response to the reviewers' comments. Your revised manuscript is also likely to be sent to reviewers for further evaluation.

Sincerely,

Mitchell F. Balish, Ph.D.

Academic Editor

PLOS Pathogens

Michael Wessels

Section Editor

PLOS Pathogens

Kasturi Haldar

Editor-in-Chief

PLOS Pathogens

orcid.org/0000-0001-5065-158X

Michael Malim

Editor-in-Chief

PLOS Pathogens

orcid.org/0000-0002-7699-2064

Three reviewers and I have evaluated the manuscript, and as you can see, there was considerable disagreement about whether the work represents a sufficiently great advance to warrant publication in PLOS Pathogens. I find that it potentially does, but there are some important matters to address concerning the description of the results and their interpretation. I agree with Reviewer 1 that although these static images could relate to the mechanism of motility, there isn't sufficient evidence to be quite as conclusive about that connection as written in the Discussion section. Overall, in addition to addressing the suggestions of the reviewers and my own comments below, the limitations of this work (including the absence of substrate in the structures) should be made explicitly clear, and how this work is a significant advance from other published work on both the M. genitalium and M. pneumoniae structures should be described. With all of that laid out carefully, it will become much clearer whether this work reaches the bar for publication in PLOS Pathogens.

The word "nap" has been misused in recent manuscripts on the adherence complexes of Mycoplasma genitalium and Mycoplasma pneumoniae, and your next paper - whether it's this one or another - is a great opportunity to rectify the usage. "Nap" was never meant to stand in as the name of an adherence-associated complex of proteins. The word "nap," which was first used in regard to M. pneumoniae in 1982 (Hu et al., Science, 216:313-315), comes from the textile industry. It means "the raised hairs, threads, or similar small projections on the surface of fabric or suede" (from Oxford Languages). It is a collective noun, like "dirt." It refers to the appearance of the apparently raised material on the surface of the attachment organelle. Just like for dirt, we wouldn't call a single particle of dirt "a Dirt." We would call it "a dirt particle." If "P140/P110 complex" isn't sufficiently concise, I recommend calling this individual tetramer a "nap particle" - all lowercase, as proteins, chemicals, and other non-living biochemical entities are not capitalized in English.

One of the reviewers had these suggestions to me, which I'll pass along to you:

• 4 times a comma is missing, either upstream or downstream of “respectively“, pls also check commas when starting with “Subsequently“ or “Initially“

• Missing blanks at “Figure 1 a&c“/“Figure 1 g,h“..(no &?)/“Figure 2g“/“Figure 2h“

• lys1289 and lys818 should read Lys1289 and Lys818

• “the pneumoniae cluster“ should read “the pneumoniae cluster“ (no species->no italics)

• “..inhibit adhesion (Vizarraga 2020)“ should read “..inhibit adhesion (26).“

• “CO2“ should read “CO2“

• “Großlöblichau“ should read “Großlöbichau“

• “Clearly misaligned particles…were removed for further processing“: really? Or were they instead excluded from further processing?

• “…box size of 300 pixel“ should read “…box size of 300 pixels“

And I'll add some of my own:

• First line page 6 – change “are slightly wider apart” to “is slightly greater”

• Next to last paragraph of discussion – P30 is M. pneumoniae; P32 is M. genitalium

• Make sure you've written "M. pneumoniae" and not "M. pneumonia" in all instances

Reviewer's Responses to Questions

**Part I - Summary**

Reviewer #1: The manuscript by Sprankel et al focuses on Nap, a protein complex responsible for cytoadherence of Mycoplasma genitalium, and reveals the structural polymorphism of intact Nap on the cell surface by cryo-ET. Although the mechanism of Nap binding and release state with sialylated oligosaccharides is an important issue, this manuscript does not provide direct evidence that the ‘open’ and ‘closed’ conformation is involved in the mechanism. Moreover, the present manuscript represents more limited progress compared to the previously published paper on the structure of Nap published by the same laboratory. Extensive additional experiments are needed to draw convincing conclusion in this study.

Reviewer #2: I read this body of work with interest. While am no expert in structural biology it is my opinion this is a rigorous piece of work that sheds new light on the mechanism of NAP function. I could not find any major drawbacks in experimental design.

Reviewer #3: In their manuscript „Cryo-electron tomography reveals the binding and release states of the native Nap adhesion complex“, the authors Sprankel, Scheffer, Ermel & Frangakis report new insights into the major adhesion complex of Mycoplasma genitalium, the Nap complex.

My thoughts after reading the manuscript, briefly summarised:

It has been a long way from the first reports in 1984 about the special flask-shaped morphology of some motile Mycoplasmas (including M. genitalium) exhibited some undefined nap structures on the bacterial surface, to the deciphering of the amino acid sequence of adherence-mediating sites of the MgPa protein in 1992, to elucidating the transcriptonal linkage of P140 and P110 in 2003, and ending now in the description of the tetrameric conformation of the nap structure in its native state at an nice 11 Ångström resolution. What a nice piece of progress over time, and this manuscript definitively is part of the journey …

The authors once again proved the many interactions between P140 (also once called MgPa or MgpB or MG191) and P110 (or MgpC or MG192), forming the heterodimers, and how they together create the pocket for binding sialic acid receptors, making it obvious why these gene products are transcriptionally linked. In addition, the authors proved again the importance of studying molecules in its native state when one considers their now presented findings about the four-helices bundles of the intracellular regions of P140/P110, and that the tetrameric organization as well as the intracellular domains (and eventually other cytoplasmic proteins) are required for the correct conformation of the extracellular globular domains and the receptor binding site….. and that a “small difference in the distance between the stalk domains translates into a large difference in the crown region“ and thereby explaining how a small, distinct intracellular impetus may have a big effect at the extracellular binding site.

For me, this manuscript is clearly worth publishing.

**Part II – Major Issues: Key Experiments Required for Acceptance**

Reviewer #1: 1. The authors’ elucidation of the ‘open’ and ‘closed’ conformations of the Nap structure at a higher resolution than before is intriguing. However, direct evidence linking this structural variation to the function of Nap is lacking. The authors should demonstrate the structural difference in the presence of sialylated oligosaccharides, rather than in a solvent without the binding target. The present data lacks sufficient impact as the previous study that have already been reported on the comparisons between closed and open states. To support the authors’ assertion, it would be necessary to showcase the structural polymorphism of Nap during both binding and release state with sialylated oligosaccharides.

2. Although this manuscript demonstrates a large amount of data, the authors should be more cautious in their conclusions and avoid giving the impression that his work provides definitive evidence for binding and release state throughout the manuscript.

3. It is worth nothing that the observation of the structure of the intracellular regions is a notable achievement. However, no discernible differences were observed in this region between the open and closed states. Therefore, it is advisable to minimize the discussion regarding intracellular interactions.

Reviewer #2: I have none

Reviewer #3: No major issues...

**Part III – Minor Issues: Editorial and Data Presentation Modifications**

Reviewer #1: In the Discussion section, authors mention that “Evidence for this is the fact that mycoplasma cells with a decreased density of Naps on the cell surface show faster gliding velocity…” Please provide the appropriate reference supporting this statement.

This manuscript lacks the appropriate references to other related papers.

Reviewer #2: The manuscript is well written and I have no issue with style or grammar. I found the manuscript easy to read.

Reviewer #3: (No Response)

PLOS authors have the option to publish the peer review history of their article (what does this mean?). If published, this will include your full peer review and any attached files.

Reviewer #1: No

Reviewer #2: **Yes: **Steven Djordjevic

Reviewer #3: No
---

## [Decision Letter · Decision Letter 1]

17 Oct 2023

Dear Dr. Frangakis,

We are pleased to inform you that your manuscript 'Cryo-electron tomography reveals the binding and release states of the major adhesion complex from Mycoplasma genitalium' has been provisionally accepted for publication in PLOS Pathogens.

Best regards,

Mitchell F. Balish, Ph.D.

Academic Editor

PLOS Pathogens

Michael Wessels

Section Editor

PLOS Pathogens

Kasturi Haldar

Editor-in-Chief

PLOS Pathogens

orcid.org/0000-0001-5065-158X

Michael Malim

Editor-in-Chief

PLOS Pathogens

orcid.org/0000-0002-7699-2064

Reviewer Comments (if any, and for reference):

Reviewer's Responses to Questions

**Part I - Summary**

Reviewer #1: The manuscript is greatly improved. My concerns have all been addressed.

Reviewer #2: (No Response)

Reviewer #3: This is a slightly improved version of what I had previously summarized as "clearly worth publishing".....and I still stand by that point of view!

**Part II – Major Issues: Key Experiments Required for Acceptance**

Reviewer #1: (No Response)

Reviewer #2: (No Response)

Reviewer #3: No more experiments required!

**Part III – Minor Issues: Editorial and Data Presentation Modifications**

Reviewer #1: (No Response)

Reviewer #2: (No Response)

Reviewer #3: All minor editorial aspects were addressed properly.

PLOS authors have the option to publish the peer review history of their article (what does this mean?). If published, this will include your full peer review and any attached files.

Reviewer #1: No

Reviewer #2: **Yes: **Steven Djordjevic

Reviewer #3: No

---

## [Editor Report · Acceptance letter]

2 Nov 2023

Dear Dr. Frangakis,

We are delighted to inform you that your manuscript, "Cryo-electron tomography reveals the binding and release states of the major adhesion complex from Mycoplasma genitalium," has been formally accepted for publication in PLOS Pathogens.

Best regards,

Kasturi Haldar

Editor-in-Chief

PLOS Pathogens

orcid.org/0000-0001-5065-158X

Michael Malim

Editor-in-Chief

PLOS Pathogens

orcid.org/0000-0002-7699-2064